# Differential Study of Retinal Thicknesses in the Eyes of Alzheimer’s Patients, Multiple Sclerosis Patients and Healthy Subjects

**DOI:** 10.3390/biomedicines11123126

**Published:** 2023-11-24

**Authors:** Elena Garcia-Martin, Daniel Jimeno-Huete, Francisco J. Dongil-Moreno, Luciano Boquete, Eva M. Sánchez-Morla, Juan M. Miguel-Jiménez, Almudena López-Dorado, Elisa Vilades, Maria I. Fuertes, Ana Pueyo, Miguel Ortiz del Castillo

**Affiliations:** 1Department of Ophthalmology, Miguel Servet University Hospital, 50009 Zaragoza, Spain; elisavilades@hotmail.com (E.V.); misabelfuertes@gmail.com (M.I.F.); anapueyobestue@gmail.com (A.P.); 2Miguel Servet Ophthalmology Innovation and Research Group (GIMSO), Aragon Institute for Health Research (IIS Aragon), Biotech Vision SLP (Spin-Off Company), University of Zaragoza, 50009 Zaragoza, Spain; 3Biomedical Engineering Group, Department of Electronics, University of Alcalá, 28871 Alcalá de Henares, Spain; d.jimeno@uah.es (D.J.-H.); fjavier.dongil@fgua.es (F.J.D.-M.); jmanuel.miguel@uah.es (J.M.M.-J.); almudena.lopez@uah.es (A.L.-D.); 4Institute of Psychiatry and Mental Health, Hospital General Universitario Gregorio Marañón, 28007 Madrid, Spain; 5School of Medicine, Universidad Complutense, 28040 Madrid, Spain; 6School of Physics, University of Melbourne, Melbourne, VIC 3010, Australia; migueloc41@gmail.com

**Keywords:** multiple sclerosis, Alzheimer’s disease, optical coherence tomography, posterior pole

## Abstract

Multiple sclerosis (MS) and Alzheimer’s disease (AD) cause retinal thinning that is detectable in vivo using optical coherence tomography (OCT). To date, no papers have compared the two diseases in terms of the structural differences they produce in the retina. The purpose of this study is to analyse and compare the neuroretinal structure in MS patients, AD patients and healthy subjects using OCT. Spectral domain OCT was performed on 21 AD patients, 33 MS patients and 19 control subjects using the Posterior Pole protocol. The area under the receiver operating characteristic (AUROC) curve was used to analyse the differences between the cohorts in nine regions of the retinal nerve fibre layer (RNFL), ganglion cell layer (GCL), inner plexiform layer (IPL) and outer nuclear layer (ONL). The main differences between MS and AD are found in the ONL, in practically all the regions analysed (AUROC_FOVEAL_ = 0.80, AUROC_PARAFOVEAL_ = 0.85, AUROC_PERIFOVEAL_ = 0.80, AUROC__PMB_ = 0.77, AUROC_PARAMACULAR_ = 0.85, AUROC_INFERO_NASAL_ = 0.75, AUROC_INFERO_TEMPORAL_ = 0.83), and in the paramacular zone (AUROC_PARAMACULAR_ = 0.75) and infero-temporal quadrant (AUROC_INFERO_TEMPORAL_ = 0.80) of the GCL. In conclusion, our findings suggest that OCT data analysis could facilitate the differential diagnosis of MS and AD.

## 1. Introduction

Alzheimer’s disease (AD) and multiple sclerosis (MS) are the two most prevalent neurological diseases today. Worldwide, 32.3 million people are currently estimated to have AD dementia [1], while in 2020, the number of MS patients worldwide stood at 2.8 million [2]. Both AD and MS are diseases with multi-modal symptomatology: they are correlated with cognitive scores, neuropathology vital signs, symptoms, demographics, medical history, neuropsychological battery, lab tests, etc. As these diseases have no cure, early diagnosis is essential for effective disease management and to optimize patient outcomes.

Currently, there is no single marker for the diagnosis of these two neurodegenerative diseases. They both require a battery of tests such as magnetic resonance imaging (MRI) and lumbar puncture. Other available tests, such as single-photon emission tomography (SPECT) and positron emission tomography (PET), likewise entail an element of risk due to their use of radioactive markers. In some cases, these diagnostic options are restricted or delayed by high costs and limited availability.

AD is primarily detected by means of four types of biomarkers [3]: brain imaging/neuroimaging (structural, functional, molecular), cerebrospinal fluid proteins, blood and urine tests and genetic risk profilers. MS is currently diagnosed via the updated McDonald criteria [4], which are based on detecting temporal and spatial alterations using MRI and clinical criteria.

The retina is an extension of the central nervous system (CNS) and is accessible via non-invasive techniques such as optical coherence tomography (OCT). The retinal nerve fibre layer (RNFL), mainly composed of axons, and the inner plexiform layer (IPL), mainly composed of cell bodies and dendrites, are not sheathed in meninges or myelin at the level of the retina. This means that OCT analysis of the thickness of these structures allows practitioners to quantify axonal damage and monitor it over time [5].

Post-mortem studies observe alterations in the retina, especially in the form of axonal degeneration and degeneration of the ganglion cell layer (especially M-type cells), in both MS patients [6] and AD patients [7].

OCT enables practitioners to observe thinning of the peripapillary retinal nerve fibre layer (pRNFL) in MS patients, and this thinning correlates with cognitive and physical disability in persons with this disease [8]. Numerous authors have also found thinning of the macular ganglion cell layer (GCL) and the inner plexiform layer (IPL) in these patients [9,10].

Most recent studies comparing control subjects with AD patients demonstrate the existence of RNFL thinning in both the pRNFL (overall and predominantly in the superior and inferior sectors [11,12,13]) and in the inner retinal layers at the level of the macula: mRNFL, GCL and IPL [11,14]. While studies detecting alterations in the pRNFL are the most numerous, other papers consider that measuring the macula may be more effective in assessing neurodegenerative changes [15]. For example, in [16], the average, superior and inferior quadrant pRNFL, mRNFL, GCL and IPL thicknesses were significantly decreased in the AD group versus controls; superior quadrant pRNFL thickness was positively associated with MMSE (Mini-Mental State Examination) scores; and macular retinal thickness exhibited absolute superiority in AD diagnosis supported by artificial intelligence.

As evidenced, in MS and AD, the thinning of certain layers of the retina presents around the optic disc and in the macula. However, these findings are not specific to these diseases as they also occur in other neurological pathologies [17,18]. It is therefore of value to acquire new insight regarding the possible differences between the two diseases. 

In this context, the purpose of this paper is to analyse the differences in retinal structure in both patient types in order to increase insight into the two diseases and provide more specific information about retinal degeneration in MS and AD. Our study aims to demonstrate the utility of OCT not only in the diagnosis of patients presenting MS or AD, but also as a possible tool for the differential diagnosis of these two neurodegenerative pathologies via an innocuous, inexpensive and easily performed technique. 

## 2. Materials and Methods

### 2.1. Study Cohort

Three independent samples of subjects aged ≥ 60 years, one comprising relapsing–remitting multiple sclerosis (RRMS) patients, one comprising AD patients and the other comprising healthy control (HC) subjects, were prospectively recruited from three clinics (an ophthalmology clinic specializing in neuro-ophthalmology and neurology clinics specializing in demyelinating diseases and dementias).

Based on a preliminary study of MS patients conducted by our group [19], we computed the sample size needed to detect differences of at least 6 μm in OCT-measured thicknesses. We used a bilateral test with an α 5% risk and a β 10% risk, i.e., with a power of 90%. To obtain enough patients for an in-depth study of the natural history of MS, it was decided to have equal numbers of non-exposed and exposed patients (ratio of 0.5). Based on these calculations, it was concluded that at least 12 eyes were needed in each group. Standard clinical and neuroimaging criteria were used as a basis for the definitive diagnosis of MS [4]. To ensure a homogeneous population, only patients with the RRMS phenotype and without a history of optic neuritis in either eye were included.

To diagnose idiopathic AD, information was collected on the participants’ medical history, their relatives’ medical history and their clinical presentation. A study of the presence of neurological and neuropsychological features, combined with advanced medical imaging techniques, was also performed to rule out brain diseases, types of dementia or other alternative medical conditions [20,21]. The healthy control group had no history of eye or neurological disease and showed no signs or symptoms of either.

Criteria for exclusion from the study were poor vision (corrected visual acuity < 0.5), large refractive errors (>5 dioptres of spherical equivalent refraction or 3 dioptres of astigmatism), high intraocular pressure (>20 mmHg), media opacification (nuclear colour/opalescence, cortical or posterior subcapsular lens opacity <2 according to the Lens Opacities Classification System III) [22] or certain eye diseases (such as glaucoma or retinal pathology). Participants with other systemic conditions that could affect the visual system were also excluded. Prior to starting the study, participants underwent full ophthalmological examination, including review of their medical history, a test of visual acuity, biomicroscopy of the anterior segment using a slit lamp, Goldmann applanation tonometry and ophthalmoscopy of the posterior segment. OCT measurements of the neuroretinal structure were also taken in all cases.

The study was conducted in accordance with the Declaration of Helsinki, and all participants provided written informed consent. The study protocol was approved by the Clinical Research Ethics Committee of Aragon (Zaragoza, Spain).

### 2.2. OCT Method

The thicknesses of the retinal layers were measured using a Spectralis spectral domain OCT device (Heidelberg Engineering) with the OCT2 Module and the Posterior Pole Retina Thickness Map (PPOLE) protocol. This protocol scans a 30 × 25° macular cube centred on the fovea (~8.8 × 7.4 mm). This automatically obtains the line that connects the centre of the fovea and the centre of the optic disc (Figure 1). This avoids differences in cell inclination depending on the subject’s chin position during the test. It also ensures that the papillomacular beam is precisely centred and that in future scans of the same patient, the same anatomical area will be analysed. In parallel to obtaining the line, the protocol explores 61 B-scans, each comprising 768 A-scans (123 microns between B-scans and 10 frames averaged per B-scan location). The thicknesses obtained are presented in 64 cells, each measuring 3° × 3°, distributed in 8 rows and 8 columns (Figure 1) [23]. 

The current commercial OCT2 Module has a scanning speed of 85,000 A-scans/s and a central wavelength of 880 nm. The scan depth is 1.9 mm, the axial resolution is 3.87 μm and the lateral resolution is 5.7 μm [24].

First-generation OCT equipment [25] worked in the time domain and required a movable mechanical reference mirror. The acquisition speed was very limited (typically 2000 A-scans/s). Later generations—spectral domain OCT (SD-OCT) and swept-source OCT (SS-OCT)—use Fourier transform analysis, avoiding mechanical movements and consequently considerably increasing the acquisition speed and improving the signal-to-noise ratio [26]. In SS-OCT, a broadband swept source whose wavelength varies with time is used. Although the SS-OCT acquisition speed is greater than that of SD-OCT (~100,000 versus ~85,000 A-scans/s) and its use of longer wavelengths (1060 nm versus 840–850 nm in SD-OCT) achieves greater tissue penetration, SD-OCT equipment is widely available and found in a large number of hospitals [27].

The device’s embedded segmentation software obtains the thickness of each of the 64 cells and provides the measurements of 9 structures or layers. This study presents data for the most representative layers, namely those of the neural retina, as previous studies have observed that the other layers have no clinical significance in these neurodegenerative pathologies [28]. These layers are the retinal nerve fibre layer (RNFL), ganglion cell layer (GCL), inner plexiform layer (IPL) and outer nuclear layer (ONL; contains cell bodies of rods and cones).

All scans were performed by the same experienced operator. There was a time delay between scan acquisition sessions, and subject position and focus were randomly altered so that the alignment parameters had to be readjusted at the start of each image acquisition session [19]. 

Images were acquired using image alignment eye-tracking TruTrack (software 7.04, Heidelberg Engineering, Heidelberg, Germany). An internal fixation target was used because this method is reported to have the highest reproducibility. No manual correction was applied to the OCT output. The quality of the scans was assessed prior to analysis, and poor-quality scans were rejected. The Spectralis OCT device uses a quality score range between 0 (poor quality) and 40 (excellent quality). Only images that scored higher than 25 were analysed. Images with artefacts, missing parts or with seemingly distorted anatomy were excluded and these scans were repeated. MS patient image quality was checked against the OSCAR-AI guidelines [29]. 

### 2.3. Study of Differences in AUC

The area under the receiver operating characteristic (AUROC) curve was used to evaluate the possible differences in thickness between the various retinal structures scanned. If X and Y are two sets of measurements, Bamber [30] showed that AUROC = Prob(X > Y). It has also been shown that the AUROC value is equal to the non-parametric Mann–Whitney U statistic divided by n_X_ × n_Y_, with n_X_ and n_Y_ being the sample sizes. 

We consider that values of AUROC > 0.75 between two samples indicate that a significant difference exists between those samples. 

### 2.4. Analysis Regions

The results are presented as cell clusters in 3 concentric rings and in 6 zones, as defined in Figure 2.

## 3. Results

### 3.1. Participating Subjects

Three independent age-matched samples of 33 RRMS patients, 21 AD patients and 19 healthy controls were prospectively included and evaluated. The demographic and clinical characteristics of the three cohorts are shown in Table 1. It should be noted that data from newly diagnosed RRMS patients (2–36 months) are used, while the time since AD diagnosis is 0.5–3 years. There are no statistically significant differences in age, sex or ophthalmological parameters between the three groups (healthy, AD and MS).

Figure 3 shows OCT slice images of the foveal area in which the segmentation in an AD patient, in an MS patient and in a healthy subject are marked. The figure shows that the thicknesses in the healthy subject are slightly greater. The regions that correspond to the areas traced in the analysis are indicated at the top of each image.

Figure 4 shows the average thicknesses in the control, MS and AD cohorts for the RNFL, GCL, IPL and ONL. It shows that the two diseases, when analysed together, produce thinning in different layers and regions of the retina, therefore requiring more detailed analysis in order to draw conclusions.

### 3.2. Differences between Control Subjects and Patients (MS and AD)

Figure 5 and Table 2 show the AUROC values for the four layers studied in the control subjects and patients (MS and AD). The greatest discriminant capacity is observed in the GCL and IPL in practically all of the regions analysed. According to this protocol, and in our database, in the ONL, alteration is only detected in the parafoveal, perifoveal, PMB and paramacular zone, while in the RNFL, the changes detected are minimal.

The layers that suffer the greatest decrease in thickness versus the control group are those corresponding to the ganglion cells (GCL) and their synaptic and dendritic connections (IPL); there is no evidence of such marked thinning in any of the other layers. It is therefore the layers whose prolongation becomes the central nervous system (internal neural layers of the retina, those corresponding to the neurosensory retina: GCL and IPL) that present greatest affectation in RRMS and AD.

As Figure 5 and Table 2 show, the zones that optimize differential diagnosis between patients with neurodegenerative diseases and control subjects are those in the PMB, which is the zone that has the greatest density of nerve fibres associated with central vision, and particularly those in the innermost layers of the retina (especially the GCL and IPL).

### 3.3. Differences between Control Subjects and MS Patients

Figure 6 and Table 3 show the differences between control subjects and RRMS patients. Although no significant alterations are found in the RNFL, they are found in most of the analysis regions in the GCL, IPL and ONL.

### 3.4. Differences between Control Subjects and AD Patients

The difference between control subjects and AD patients, as evaluated using the AUROC, is shown in Figure 7 and Table 4. In this comparison, the differences are mainly found in the GCL and IPL. These layers appear to be useful for AD diagnosis as they provide a means of identifying control subjects. In addition, analysis reveals that in both layers, the zone with greatest discriminant capacity is the central zone of the PMB between the optic nerve and the macula. The superior zone appears to contain information of greater relevance in discriminating between AD patients and control subjects than the inferior zone.

### 3.5. Differences between MS and AD Patients

Comparing the two patient cohorts with each other produces the results in Figure 8 and Table 5; the differences are manifested in the ONL (three rings, PMB, paramacular, and IN and IT quadrants) and in the GCL (paramacular and IT quadrant).

To illustrate the thickness differences in the ONL in greater detail, Figure 9 presents their distribution in the three cohorts and in the nine analysis regions. It corroborates that thinning does not generally occur in AD, while it does occur in MS.

## 4. Discussion and Conclusions

The literature to date encompasses extensive reviews that elucidate changes in retinal thickness in both MS [8,10] and AD [11,12]. To the best of our knowledge, however, there are no prior studies that compare the differential diagnostic capacity of OCT in relation to these two diseases. This paper therefore provides evidence that the thickness of the retina could provide valuable insights into the differential diagnosis of MS and AD.

The results presented in this paper confirm previous studies relating to the affectation of the retinal structure in MS and AD. In MS patients, we observed that the zones with the highest AUROC values are consistent with our earlier research, detecting thinning mainly in the internal layers of the retina (GCL and IPL). In terms of topographical distribution in the retina, the cells exhibiting greatest thinning are those corresponding to the PMB, which is consistent with previous studies of OCT and MS that find that most thinning occurs in these zones [31,32].

Our findings also indicate significant thinning in the ONL in MS patients, which may be associated with the loss of cones that occurs over the course of this disease [33].

The results of this paper match those of many of the studies analysing changes in thickness due to AD. Our results show that the RNFL does not exhibit large variations in AD patients versus controls, which is consistent with the meta-analysis [12], which revealed a small range of significance in this layer. While some studies in AD patients have found remarkable thickening of the ONL [34,35], others do not reach the same conclusion [36] when comparing them with control subjects. In our paper, although we detected a slight alteration in the ONL versus control subjects, the highest AUROC value was 0.67 (SN quadrant). 

We did, however, observe that the internal retinal layers (GCL, IPL) showed the greatest discriminant capacity in practically all of the Posterior Pole protocol cells, in line with the conclusions drawn in studies such as [11,14], which report significant differences in GC–IPL thickness in AD patients.

As Figure 8 and Table 5 show, when performing a differential diagnosis between MS and AD, practitioners must primarily examine the ONL, as the data reveal that pronounced thinning of this layer occurs in MS, while in AD patients, ONL thicknesses are very similar to those in the control group (Figure 9). As shown in Figure 8, the greatest differences are found in the IT quadrant and in the paramacular and parafoveal zones. Comparing MS and AD also reveals differences in the GCL (albeit to a lesser extent), in the paramacular zone and in the IT quadrant.

What we therefore observe in this study is that each disease has a different topographical distribution or, expressed in other words, each pathology has its own fingerprint. This is consistent with the findings of our research group and those of others studying the benefits of using OCT to measure affectation in MS [10,37] and AD patients [38].

According to our findings, the ONL appears to be key to differential diagnosis between MS and AD as it is more affected in MS. This layer contains several strata of oval nuclear bodies (rod and cone granules, so named on account of their being, respectively, connected to the rods and cones of the next layer, the photoreceptor layer). One possible explanation for this thinning that occurs in MS and not in the other neurodegenerative disease studied may be that eyes affected by RRMS suffer inflammatory processes such as optic neuritis that cause a thinning of the outer layers of the retina, as well as a predominant phenotype of macular thinning that does not produce significant thinning of the RNFL [39]. These studies raise the question of whether the damage caused by MS can spread to the outer layers of the retina. In addition, increases in the thickness of the INL and of the microcystic oedema associated with the development of new lesions have been identified in MS patients using contrast enhancement and T2 [40]. However, few prospective studies have been conducted to elucidate the changes in the thickness of the outer retinal layers that occur over time in the context of acute optic neuritis and RRMS [41]. The mechanical and pressure-related processes that occur in RRMS and that thin the ONL do not occur in AD, probably because it is a purely neurodegenerative disease and does not present alongside inflammation of the optic nerve.

The strengths of this study are that all the OCT recordings were taken by the same operator using the same protocol, thereby ensuring that the procedure was highly homogeneous. The patients had received a definitive diagnosis from an expert neurologist, which allows the exclusion of other types of dementia or neurological or neurodegenerative disease that could cause additional alteration to the neuroretinal structure.

The main weakness of this study is the small database due to the difficulty of obtaining MS and AD recordings for age-matched populations. This is significant because it is important that both groups are similar in age so that there is no additional retinal layer loss due to ageing. Another potential limitation, controlled in this study through the inclusion and exclusion criteria, is the presence of ocular disturbances such as cataracts, which could prevent gaze fixation during the exploratory protocol. In this study, the OCT images were interpreted by two neuro-ophthalmologists with extensive experience in both this field and in assessing OCT tests. 

Possible improvements to consider in future studies in this area include increasing the sample size and implementing a multi-variable approach that takes into account other clinical variables applicable to the subjects. Also, although the subjects in the AD group were not at very advanced stages, it would be interesting to conduct a study solely comprising newly diagnosed subjects or subjects at stages prior to AD (e.g., mild cognitive impairment). This would help to identify whether the alterations detected were already present in the early stages of the disease.

In conclusion, if the findings of this paper are confirmed in a broader study, this research will have identified a biomarker for MS and AD that meets all the criteria necessary to be considered efficient [42]: simple to perform, reliable, minimally invasive, inexpensive and able to detect features of the active pathophysiological processes. This is an exploratory study in relation to the diagnosis of two common diseases, using neuro-retinal thickness assessment via the innocuous OCT test. This test can be considered a useful technique that, in synergy with other diagnostic methods, helps one to reach a definitive diagnosis earlier and more efficiently.

## Figures and Tables

**Figure 1 biomedicines-11-03126-f001:**
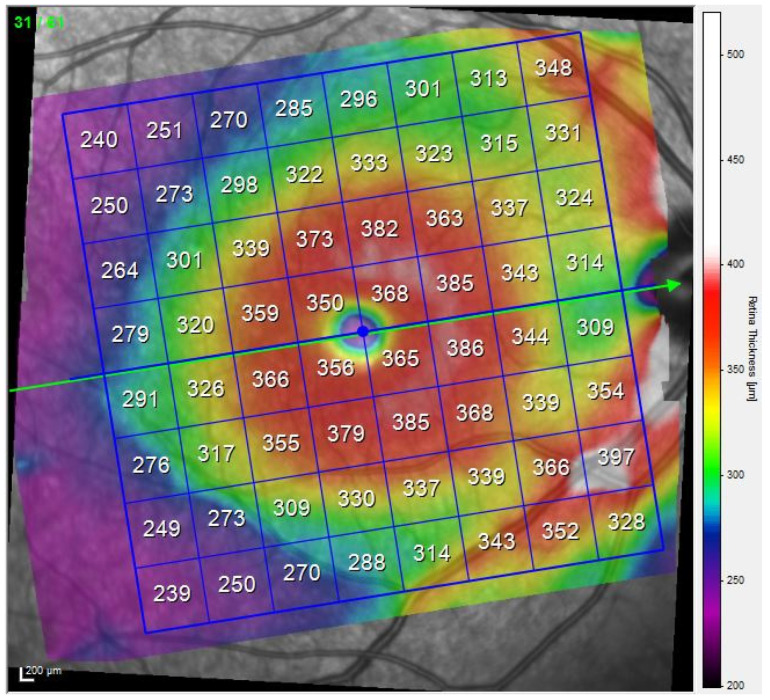
An 8 × 8 grid generated using the Posterior Pole protocol in a Spectralis OCT device. The retina, blood vessels and optic nerve (right side of the figure) are visible in the background. The grid shows the 64 cells and the average retinal thicknesses in microns for each of them. The image is of a right eye.

**Figure 2 biomedicines-11-03126-f002:**
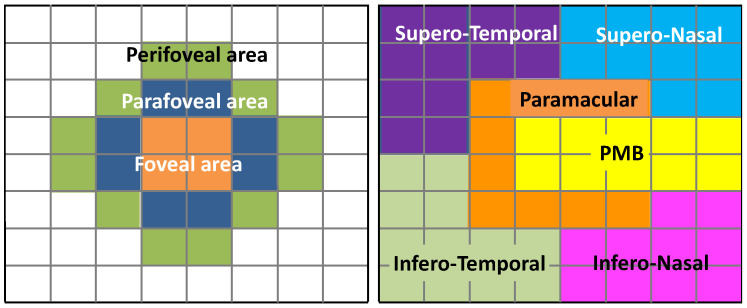
Definition of the analysis regions in the 8 × 8 array: 3 concentric rings (foveal, parafoveal, perifoveal) and 6 zones: papillomacular nerve fibre bundle (PMB), paramacular, supero-nasal quadrant (SN), infero-nasal quadrant (IN), infero-temporal quadrant (IT) and supero-temporal quadrant (ST).

**Figure 3 biomedicines-11-03126-f003:**
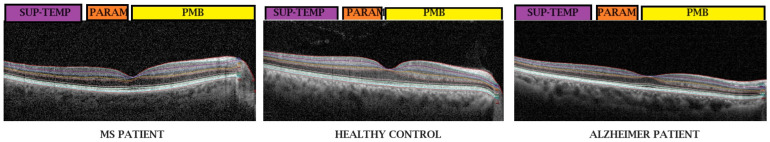
OCT slice image of the foveal area.

**Figure 4 biomedicines-11-03126-f004:**
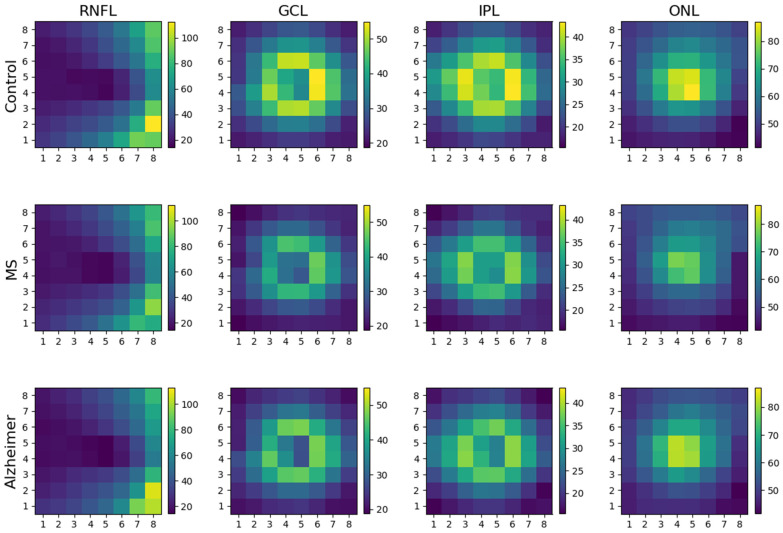
Average values *in microns* of the RNFL, GCL, IPL and ONL thicknesses in the control, MS patient and AD patient cohorts. The infero-temporal cell is positioned at coordinate (1, 1) and the supero-nasal cell is positioned at (8, 8).

**Figure 5 biomedicines-11-03126-f005:**
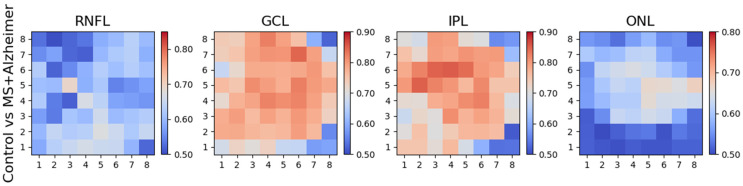
AUROC values: controls vs. patients (MS and AD).

**Figure 6 biomedicines-11-03126-f006:**
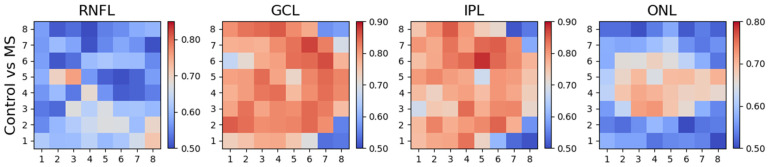
AUROC values: controls vs. MS patients.

**Figure 7 biomedicines-11-03126-f007:**
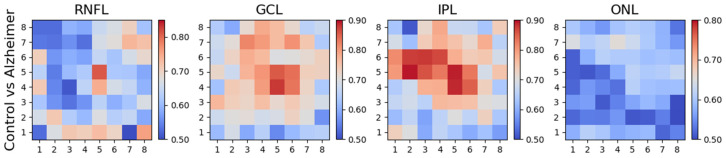
AUROC values: controls vs. AD patients.

**Figure 8 biomedicines-11-03126-f008:**
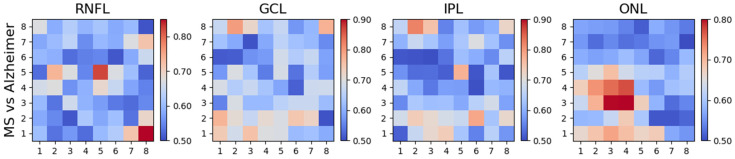
AUROC values: MS vs. AD.

**Figure 9 biomedicines-11-03126-f009:**
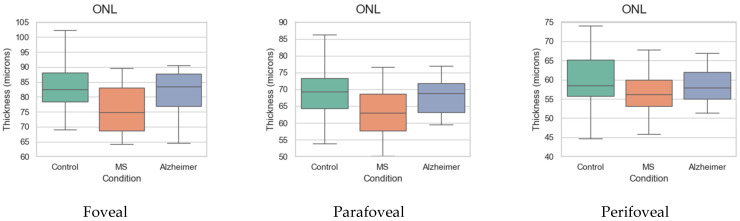
Box plots of interquartile range (IQR), range and median ONL thickness in different analysis regions for ONL layer.

**Table 1 biomedicines-11-03126-t001:** Demographic and clinical characteristics of the multiple sclerosis (MS) patients, Alzheimer disease (AD) patients and healthy controls. Abbreviations: SD, standard deviation; EDSS, expanded disability status scale; MMSE, mini-mental state examination.

	MS Patients (*n* = 33 Patients)	AD Patients (*n* = 21 Patients)	Healthy Controls (*n* = 19 Subjects)	*p*-Value
Age (years) (mean (SD))	69.53 (6.81)	69.73 (5.05)	69.10 (9.25)	*p* = 0.917
Male/female ratio	3/30	9/12	9/12	*p* = 0.165
Visual acuity (Snellen)	0.83 (0.12)	0.79 (0.14)	0.84 (0.15)	*p* = 0.333
Spherical refractive error (dioptres)	−1.45 (0.21)	−1.24 (0.34)	−1.53 (0.19)	*p* = 0.122
Axial length (mm)	22.13 (2.43)	22.54 (3.01)	22.43 (2.09)	*p* = 0.448
Disease duration since definitive diagnosis (years) (mean (SD))	1.35 (0.66)	1.46 (0.93)	---	---
EDSS score (median (range))	1.31 (0–3.5)	---	---	---
MMSE score (mean (SD))	----	19.21 (3.76)	----	
Treatment	Avonex: 8 Betaseron: 7 Rebif: 5 Glatiramer acetate: 2 Tecfidera: 1 Gilenya: 2 Mayzent: 1 Aubagio: 6 No treatment: 1		---	---

**Table 2 biomedicines-11-03126-t002:** Controls vs. patients (MS and AD). AUROC values, with 95% confidence interval.

Layers	Foveal Area	Parafoveal Area	Perifoveal Area	PMB	Paramacular	SN	IN	IT	ST
RNFL	0.62 [0.51–0.77]	0.68 [0.54–0.82]	0.62 [0.51–0.75]	0.62 [0.51–0.77]	0.69 [0.56–0.82]	0.5 [0.5–0.66]	0.65 [0.52–0.8]	0.6 [0.5–0.76]	0.59 [0.51–0.72]
GCL	0.89 [0.81–0.95]	0.93 [0.87–0.97]	0.91 [0.84–0.97]	0.94 [0.89–0.99]	0.93 [0.86–0.98]	0.87 [0.78–0.94]	0.83 [0.71–0.92]	0.77 [0.65–0.88]	0.78 [0.66–0.88]
IPL	0.85 [0.76–0.93]	0.91 [0.83–0.96]	0.9 [0.81–0.96]	0.92 [0.85–0.97]	0.92 [0.84–0.97]	0.78 [0.67–0.88]	0.72 [0.59–0.84]	0.68 [0.53–0.82]	0.82 [0.7–0.92]
ONL	0.73 [0.61–0.85]	0.78 [0.64–0.89]	0.77 [0.63–0.88]	0.78 [0.64–0.89]	0.77 [0.65–0.88]	0.73 [0.58–0.86]	0.67 [0.53–0.82]	0.63 [0.51–0.78]	0.72 [0.59–0.83]

**Table 3 biomedicines-11-03126-t003:** AUROC values: controls vs. MS patients, with 95% confidence interval.

Layers	Foveal Area	Parafoveal Area	Perifoveal Area	PMB	Paramacular	SN	IN	IT	ST
RNFL	0.59 [0.51–0.76]	0.67 [0.53–0.81]	0.62 [0.51–0.76]	0.61 [0.51–0.76]	0.7 [0.55–0.83]	0.52 [0.5–0.69]	0.7 [0.56–0.83]	0.6 [0.51–0.75]	0.58 [0.5–0.74]
GCL	0.93 [0.84–0.99]	0.96 [0.9–1.0]	0.93 [0.85–0.99]	0.95 [0.88–1.0]	0.96 [0.9–1.0]	0.87 [0.78–0.96]	0.84 [0.73–0.94]	0.86 [0.75–0.95]	0.81 [0.7–0.92]
IPL	0.88 [0.77–0.96]	0.94 [0.87–0.99]	0.93 [0.85–0.99]	0.94 [0.87–0.99]	0.94 [0.88–0.99]	0.75 [0.62–0.87]	0.74 [0.61–0.87]	0.75 [0.6–0.87]	0.82 [0.7–0.93]
ONL	0.83 [0.72–0.93]	0.87 [0.76–0.95]	0.87 [0.76–0.95]	0.86 [0.74–0.95]	0.88 [0.78–0.96]	0.76 [0.61–0.88]	0.76 [0.61–0.89]	0.74 [0.57–0.88]	0.79 [0.65–0.9]

**Table 4 biomedicines-11-03126-t004:** AUROC values: controls vs. AD patients, with 95% confidence interval.

Layers	Foveal Area	Parafoveal Area	Perifoveal Area	PMB	Paramacular	SN	IN	IT	ST
RNFL	0.67 [0.52–0.83]	0.69 [0.52–0.84]	0.61 [0.51–0.79]	0.65 [0.51–0.83]	0.68 [0.52–0.84]	0.52 [0.5–0.73]	0.58 [0.51–0.76]	0.6 [0.5–0.78]	0.6 [0.51–0.76]
GCL	0.84 [0.7–0.95]	0.89 [0.77–0.97]	0.88 [0.76–0.97]	0.93 [0.86–0.99]	0.89 [0.78–0.98]	0.86 [0.73–0.96]	0.81 [0.64–0.94]	0.63 [0.51–0.8]	0.72 [0.55–0.88]
IPL	0.81 [0.68–0.94]	0.85 [0.71–0.95]	0.85 [0.71–0.96]	0.88 [0.76–0.97]	0.87 [0.75–0.96]	0.83 [0.68–0.95]	0.69 [0.52–0.86]	0.57 [0.5–0.76]	0.82 [0.67–0.93]
ONL	0.57 [0.5–0.75]	0.63 [0.51–0.8]	0.61 [0.5–0.78]	0.66 [0.51–0.81]	0.61 [0.51–0.79]	0.67 [0.51–0.83]	0.52 [0.5–0.71]	0.54 [0.5–0.73]	0.61 [0.51–0.78]

**Table 5 biomedicines-11-03126-t005:** AUROC values: MS vs. AD. With 95% confidence interval.

Layers	Foveal Area	Parafoveal Area	Perifoveal Area	PMB	Paramacular	SN	IN	IT	ST
RNFL	0.59 [0.51–0.74]	0.53 [0.5–0.68]	0.55 [0.5–0.7]	0.5 [0.5–0.67]	0.59 [0.5–0.74]	0.55 [0.5–0.7]	0.68 [0.54–0.82]	0.56 [0.5–0.7]	0.52 [0.5–0.68]
GCL	0.64 [0.51–0.77]	0.74 [0.6–0.86]	0.75 [0.61–0.87]	0.69 [0.54–0.83]	0.75 [0.61–0.88]	0.6 [0.5–0.74]	0.66 [0.52–0.79]	0.8 [0.67–0.9]	0.67 [0.52–0.81]
IPL	0.61 [0.5–0.76]	0.69 [0.54–0.83]	0.71 [0.57–0.84]	0.64 [0.52–0.78]	0.69 [0.54–0.83]	0.52 [0.5–0.68]	0.6 [0.51–0.76]	0.73 [0.58–0.86]	0.59 [0.51–0.72]
ONL	0.8 [0.67–0.92]	0.85 [0.73–0.93]	0.8 [0.69–0.91]	0.77 [0.63–0.9]	0.85 [0.74–0.95]	0.63 [0.51–0.78]	0.75 [0.6–0.88]	0.83 [0.69–0.94]	0.68 [0.53–0.83]

## Data Availability

The data collected and/or analysed during the current study are available from the corresponding author upon reasonable request.

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
