# Peer review of "Differential Study of Retinal Thicknesses in the Eyes of Alzheimer’s Patients, Multiple Sclerosis Patients and Healthy Subjects"

_biomedicines, 2023, doi:10.3390/biomedicines11123126_

Round 1

Reviewer 1 Report

Comments and Suggestions for Authors

Since AD and MS have different systemic clinical findings, the differential diagnosis is not difficult to make in daily clinical practice. Therefore, the usefulness of differential diagnosis focusing on changes in retinal thickness is poor.

1.       It is hard to believe that the ONL is impaired in MS and there is little logical evidence to support this. To prove this, it is necessary to perform tests such as electroretinograms (ERG). In addition, have there actually been cases where the ellipsoid zone or interdigitation zone is impaired?

2.       The relationship between each retinal layer and visual function, including visual field test and ERG, should be examined.

3.       One of the exclusion criteria is a best-corrected visual acuity less than 0.5, but why 0.5?" It is commonly set as "lower than 1.0" and if it is less than 1.0, ocular disease is suspected. The authors should clarify in more detail the rationale for setting it up that way in this study.

4.       Why is the grid in Figure 1 slanted? Are these angles different for each patient? If so, could this change the interpretation of the results?

5.       Data on visual acuity, refractive error, and axial length should also be included in Table 1.

Author Response

Since AD and MS have different systemic clinical findings, the differential diagnosis is not difficult to make in daily clinical practice. Therefore, the usefulness of differential diagnosis focusing on changes in retinal thickness is poor.

  1. It is hard to believe that the ONL is impaired in MS and there is little logical evidence to support this. To prove this, it is necessary to perform tests such as electroretinograms (ERG). In addition, have there actually been cases where the ellipsoid zone or interdigitation zone is impaired?

Author´s response:

Dear reviewer, thank you very much for your comment.

In previous studies carried out by our research group and others, analysis of electroretinography patterns has shown that transmission of nerve impulses is affected in MS patients and that this is associated with thinning of the visual pathway. This can be transferred anterogradely with degeneration, which ends up affecting the retinal layers, including the ONL.

Our findings (please see Fig. 9), detected in MS patients by OCT, show a thinning of the ONL in most of the analysed regions.

References:

  • Kleerekooper, I., Del Porto, L., Dell’Arti, L., Guajardo, J., Leo, S., Robson, A. G., ... & Holder, G. E. (2022). Pattern ERGs suggest a possible retinal contribution to the visual acuity loss in acute optic neuritis. Documenta Ophthalmologica, 145(3), 185-195.
  • Barbano, L., Ziccardi, L., Antonelli, G., Nicoletti, C. G., Landi, D., Mataluni, G., ... & Parisi, V. (2022). Multifocal Electroretinogram Photopic Negative Response: A Reliable Paradigm to Detect Localized Retinal Ganglion Cells’ Impairment in Retrobulbar Optic Neuritis Due to Multiple Sclerosis as a Model of Retinal Neurodegeneration. Diagnostics, 12(5), 1156.
  • Garcia-Martin, E., Ara, J. R., Martin, J., Almarcegui, C., Dolz, I., Vilades, E., ... & Satue, M. (2017). Retinal and optic nerve degeneration in patients with multiple sclerosis followed up for 5 years. Ophthalmology, 124(5), 688-696.
  • Rodriguez-Mena, D., Almarcegui, C., Dolz, I., Herrero, R., Bambo, M. P., Fernandez, J., ... & Garcia-Martin, E. (2013). Electropysiologic evaluation of the visual pathway in patients with multiple sclerosis. Journal of Clinical Neurophysiology, 30(4), 376-381.
  • Garcia-Martin, E., Rodriguez-Mena, D., Herrero, R., Almarcegui, C., Dolz, I., Martin, J., ... & Pablo, L. E. (2013). Neuro-ophthalmologic evaluation, quality of life, and functional disability in patients with MS. Neurology, 81(1), 76-83.

  1. The relationship between each retinal layer and visual function, including visual field test and ERG, should be examined.

Author´s response:

Dear reviewer, we thank you for your suggestion. Visual field and electroretinography tests were not included in this study as the aim was to perform differential analysis of retinal structures in control, MS and AD subjects using only optical coherence tomography.

However, our previous studies and those of other research groups have included visual field and electroretinography tests in patients with multiple sclerosis, and alterations have been detected in all three tests, with OCT proving more sensitive than functional tests (ERG and visual field) as regards detecting changes, especially longitudinal changes over time.

References:

  • Garcia-Martin, E., Ara, J. R., Martin, J., Almarcegui, C., Dolz, I., Vilades, E., ... & Satue, M. (2017). Retinal and optic nerve degeneration in patients with multiple sclerosis followed up for 5 years. Ophthalmology, 124(5), 688-696.
  • Barbano, L., Ziccardi, L., Antonelli, G., Nicoletti, C. G., Landi, D., Mataluni, G., ... & Parisi, V. (2022). Multifocal Electroretinogram Photopic Negative Response: A Reliable Paradigm to Detect Localized Retinal Ganglion Cells’ Impairment in Retrobulbar Optic Neuritis Due to Multiple Sclerosis as a Model of Retinal Neurodegeneration. Diagnostics, 12(5), 1156.

  1. One of the exclusion criteria is a best-corrected visual acuity less than 0.5, but why 0.5?" It is commonly set as "lower than 1.0" and if it is less than 1.0, ocular disease is suspected. The authors should clarify in more detail the rationale for setting it up that way in this study.

Author´s response:

Visual acuity of 1.0 is the situation in which the eye sees perfectly with great sharpness. It is difficult to find this visual acuity in healthy eyes of elderly patients or in subjects whose retina or optic nerve is aged by some non-ophthalmological pathology. Therefore, as in our previous studies, we set visual acuity < 0.5 as the acuity necessary to adequately perform the tests of the exploratory protocol and to fix the gaze with sufficient precision while the optical coherence tomography test is performed.

  1. Why is the grid in Figure 1 slanted? Are these angles different for each patient? If so, could this change the interpretation of the results?

Author´s response:

The inclination of the figure is due to the fact that the posterior pole protocol automatically detects the position of the fovea point, taking into account the absence of blood vessels in that area. Subsequently, the software draws a central line, shown in blue in Figure 1, between the optic nerve papilla and the macula. From this line, it performs the analysis in the cells shown. This avoids the cells having differing inclinations depending on how the patient positions their chin during the test. It also ensures that the papillomacular beam is precisely centred and that in future scans of the same patient the same anatomical area will be analysed. Thus, any change recorded will not be due to differences in subject head position. This is one of the novel features and improvements to the posterior pole protocol, which is able to automatically detect the fovea point.

Reference: The thicknesses of the retinal layers were measured using a Spectralis spectral domain OCT device (Heidelberg Engineering) and the Posterior Pole Retina Thickness Map (PPOLE) protocol. Asrani S, Rosdahl JA, Allingham RR. Novel software strategy for glaucoma diagnosis: asymmetry analysis of retinal thickness. Arch Ophthalmol. 2011;129(9):1205–1211. doi:10.1001/archophthalmol.2011.24221911669.

We have modified the first paragraph of Section 2.2 OCT method to clarity this point of the manuscript:

The thicknesses of the retinal layers were measured using a Spectralis spectral domain OCT device (Heidelberg Engineering) and the Posterior Pole Retina Thickness Map (PPOLE) protocol.  This protocol scans a 30 × 25° macular cube centred on the fovea (~8.8 × 7.4 mm). This automatically obtains the line that connects the centre of the fovea and the centre of the optic disc (Figure 1). This avoids differences in cell inclination depending on the subject’s chin position during the test. It also ensures that the papillomacular beam is precisely centred and that in future scans of the same patient, the same anatomical area will be analysed. In parallel to obtaining the line, the protocol explores 61 B-scans, each comprising 768 A-scans (123 microns between B-scans and 10 frames averaged per B-scan location). The thicknesses obtained are presented in 64 cells, each measuring 3° × 3°, distributed in 8 rows and 8 columns (Figure 1).

  1. Data on visual acuity, refractive error, and axial length should also be included in Table 1.

Author´s response:

Data on visual acuity, refractive error and axial length are now included in Table 1. Thank you very much for spotting this important detail.

Reviewer 2 Report

Comments and Suggestions for Authors

General

The study approaches OCT for a comparison between multiple sclerosis (MS) and Alzheimer's disease (AD) in terms of retinal thinning. The topic is of interest, the research is well done, with interesting results in terms of improving diagnosis capabilities of the technique, and the manuscript is in general well written. Therefore, the paper can be considered for publication with some improvements, as pointed out bellow.

Specific

1) The English and style are in general fine throughout the manuscript. However, some issues exist, for example please avoid one-phrase paragraphs. The manuscript should be revised from the point of view of style.

2) “Based on our preliminary studies in MS patients [23], we calculated the sample size needed to detect differences of at least 6 μm in OCT-measured thicknesses” Can you please elaborate? This value of 6 microns is pivotal for this work, therefore it must be justified, a citation is not enough.

3) Also, for the SD OCT system utilized, please provide a few more parameters, including wavelength and more important, axial and lateral resolution. These should be correlated to the above 6 microns threshold that was pointed out (in terms of which type of resolution?). Not sure that the system is capable to provide it. Many OCT systems have a 5 microns axial resolution, others are worse. If this resolution and the measuring threshold are close, errors can be huge. Please elaborate.

4) “Only images that scored higher than 25 were analysed.” Why this specific value? How does it translate in image quality parameters (for example contrast or CNR)?

5) “Images with artefacts, missing parts, or with seemingly distorted anatomy were excluded and these scans were repeated.” It would be interesting to give a few examples of good and bad/excluded images (the latter from different points of view), for others to successfully reproduce such protocols.

6) While the analysis and the statistics look fine, OCT images of MS vs AD patients must definitely be provided – as examples. Such images should be also (ideally) included in a Supplementary material in order to demonstrate the findings.

7) Referring to the above, the way the analysis was performed on regions and zones (Fig. 2) should be demonstrated on such OCT images – to demonstrate the developed/applied procedure.

8) Section 4 is too long for Conclusions, please rename it as Discussion and Conclusions.

9) Please refer in the Intro to (and complete the Refs list with) relevant titles on the OCT technology that are missing now:

- the first paper that introduced OCT:

D. Huang, E. A. Swanson, C. P. Lin, J. S. Schuman, W. G. Stinson, W. Chang, M. R. Hee, T. Flotte, K. Gregory, C. A. Puliafito, and J. G. Fujimoto, “Optical coherence tomography,” Science 254(5035), 1178-1181 (1991).

- at least a relevant review on the OCT technique, to point out advantages of FD OCT as the one used in the study:

M. A. Choma, M. V. Sarunic, C. Yang, and J. A. Izatt, “Sensitivity advantage of swept-source and Fourier-domain optical coherence tomography,” Opt. Express 11, 2183-2189 (2003).

- a state-of-the-art OCT technique for high (i.e. 2 microns) axial and lateral resolution

Cogliati A., Canavesi C., Hayes A., Tankam P., Duma V.-F., Santhanam A., Thompson K. P., and Rolland J. P., MEMS-based handheld scanning probe with pre-shaped input signals for distortion-free images in Gabor-Domain Optical Coherence Microscopy, Optics Express 24(12), 13365-13374 (2016)

etc.

In conclusion, the study has potential, but it should be completed and revised for publication.

Comments on the Quality of English Language

The English and style are in general fine throughout the manuscript. However, some issues exist, for example please avoid one-phrase paragraphs. The manuscript should be revised from the point of view of style.

Author Response

General

The study approaches OCT for a comparison between multiple sclerosis (MS) and Alzheimer's disease (AD) in terms of retinal thinning. The topic is of interest, the research is well done, with interesting results in terms of improving diagnosis capabilities of the technique, and the manuscript is in general well written. Therefore, the paper can be considered for publication with some improvements, as pointed out bellow.

Specific

  • The English and style are in general fine throughout the manuscript. However, some issues exist, for example please avoid one-phrase paragraphs. The manuscript should be revised from the point of view of style.

Author´s response:

Dear reviewer, thank you very much for your suggestion. The manuscript has been revised and sentences that are too long have been avoided.

  • Based on our preliminary studies in MS patients [23], we calculated the sample size needed to detect differences of at least 6 μm in OCT-measured thicknesses” Can you please elaborate? This value of 6 microns is pivotal for this work, therefore it must be justified, a citation is not enough.
  • Also, for the SD OCT system utilized, please provide a few more parameters, including wavelength and more important, axial and lateral resolution. These should be correlated to the above 6 microns threshold that was pointed out (in terms of which type of resolution?). Not sure that the system is capable to provide it. Many OCT systems have a 5 microns axial resolution, others are worse. If this resolution and the measuring threshold are close, errors can be huge. Please elaborate.

Author´s response:

The current commercial OCT2 Module has a scanning speed of 85000 A-scans/second and a central wavelength of 880 nm. The scan depth is 1.9 mm and the OCT device has an axial resolution of 3.87 μm and a lateral resolution of 5.7 μm.

Ref: McCann P, Hogg RE, Wright DM, McGuinness B, Young IS, Kee F, Azuara-Blanco A. Diagnostic Accuracy of Spectral-Domain OCT Circumpapillary, Optic Nerve Head, and Macular Parameters in the Detection of Perimetric Glaucoma. Ophthalmol Glaucoma 2019;2(5):336-345.  doi: 10.1016/j.ogla.2019.06.003. 

We chose a minimum of 6 microns for difference detection firstly to be more confident that the device is capable of detecting those differences and secondly to not set the detection threshold too high. In addition, previous studies have set this resolution at 6 microns, as previous OCT equipment had a resolution of 5 microns.

We have added a second paragraph in Section 2.2 OCT method of the manuscript to provide this information:

The current commercial OCT2 Module has a scanning speed of 85000 A-scans/second and a central wavelength of 880 nm. Scan depth is 1.9 mm, axial resolution is 3.87 μm and lateral resolution is 5.7 μm (McCann P et al, 2019).

  • “Only images that scored higher than 25 were analysed.” Why this specific value? How does it translate in image quality parameters (for example contrast or CNR)?

Author´s response:

During the OCT scan, the device provides a quality bar that scores the images from 0–40 points. The device itself marks the images in blue when it considers them to be of sufficient quality (i.e. when they score more than 25 out of 40 points). We chose this value for this reason as it is the value automatically determined by the device and is the one used in previous publications featuring Spectralis OCT.

Reference:

Hernandez, M., Ramon-Julvez, U., Vilades, E., Cordon, B., Mayordomo, E., & Garcia-Martin, E. (2023). Explainable artificial intelligence toward usable and trustworthy computer-aided early diagnosis of multiple sclerosis from Optical Coherence Tomography. arXiv preprint arXiv:2302.06613.

5) “Images with artefacts, missing parts, or with seemingly distorted anatomy were excluded and these scans were repeated.” It would be interesting to give a few examples of good and bad/excluded images (the latter from different points of view), for others to successfully reproduce such protocols.

Author´s response:

There are numerous articles and manuals explaining the proper procedure for excluding images with artefacts or areas of inadequate OCT quality. This is common in the practice of ophthalmology and optometry. Any OCT scanner with minimal OCT experience is able to see when an image includes artefacts or is not able to adequately mark and segment retinal layers. For this reason, we understand that it is not the purpose of this article to explain what high-quality images are, as it would make the article too long and it is something that can be easily consulted elsewhere. However, if the reviewer considers it important, we can include examples of images that we discarded for these reasons as supplementary material.

In any case, our images have been checked according to the OSCAR criteria, which are the most widely used in neuro-ophthalmological research to ensure adequate OCT image quality, as mentioned in the last paragraph of Section 2.2 OCT method:

Images were acquired using image alignment eye-tracking software (TruTrack, Heidelberg Engineering). An internal fixation target was used because this method is reported to have the highest reproducibility. No manual correction was applied to the OCT output. The quality of the scans was assessed prior to analysis and poor-quality scans were rejected. The Spectralis OCT device uses a quality score range between 0 (poor quality) and 40 (excellent quality). Only images that scored higher than 25 were analysed. Images with artefacts, missing parts, or with seemingly distorted anatomy were excluded and these scans were repeated. MS patient image quality was checked against the OSCAR-AI guidelines [29].

6) While the analysis and the statistics look fine, OCT images of MS vs AD patients must definitely be provided – as examples. Such images should be also (ideally) included in a Supplementary material in order to demonstrate the findings.

7) Referring to the above, the way the analysis was performed on regions and zones (Fig. 2) should be demonstrated on such OCT images – to demonstrate the developed/applied procedure.

Following the reviewer's suggestion, we include an OCT slice image of the foveal area showing the segmentation in an AD patient, in an MS patient and in a healthy subject (new Figure 3). The image shows that the thicknesses in the healthy subject are slightly greater. However, we note that while with the naked eye it is difficult to spot these differences, with the tool proposed in the article the algorithm detects these differences much more reliably.

In the image, we have marked the regions that would correspond to the areas traced in the analysis.

It should be noted that in each Spectralis acquisition with the subsequent Pole protocol, 25 slices, as shown in the figure, are acquired. These slices supplement the proposed analysis.

We have added Figure 3 and its legend as follows:

Figure 3 shows OCT slice images of the foveal area in which the segmentation in an AD patient, in an MS patient and in a healthy subject are marked. The figure shows that the thicknesses in the healthy subject are slightly greater. The regions that correspond to the areas traced in the analysis are indicated at the top of each image.

8) Section 4 is too long for Conclusions, please rename it as Discussion and Conclusions.

Author´s response: We have changed the title of this section.

Thank you.

9) Please refer in the Intro to (and complete the Refs list with) relevant titles on the OCT technology that are missing now:

- the first paper that introduced OCT:

  1. Huang, E. A. Swanson, C. P. Lin, J. S. Schuman, W. G. Stinson, W. Chang, M. R. Hee, T. Flotte, K. Gregory, C. A. Puliafito, and J. G. Fujimoto, “Optical coherence tomography,” Science 254(5035), 1178-1181 (1991).

- at least a relevant review on the OCT technique, to point out advantages of FD OCT as the one used in the study:

  1. A. Choma, M. V. Sarunic, C. Yang, and J. A. Izatt, “Sensitivity advantage of swept-source and Fourier-domain optical coherence tomography,” Opt. Express 11, 2183-2189 (2003).

- a state-of-the-art OCT technique for high (i.e. 2 microns) axial and lateral resolution

Cogliati A., Canavesi C., Hayes A., Tankam P., Duma V.-F., Santhanam A., Thompson K. P., and Rolland J. P., MEMS-based handheld scanning probe with pre-shaped input signals for distortion-free images in Gabor-Domain Optical Coherence Microscopy, Optics Express 24(12), 13365-13374 (2016)

etc.

Author´s response:

Dear reviewer, we consider your suggestion very appropriate and have therefore added the following paragraph (third paragraph of Section 2.2 OCT method):

First-generation OCT equipment [25] worked in the time domain and required a movable mechanical reference mirror. Acquisition speed was very limited (typically 2,000 A-scans/second).  Later generations – spectral domain OCT (SD-OCT) and swept-source OCT (SS-OCT) – use Fourier transform analysis, avoiding mechanical movements and consequently considerably increasing acquisition speed and improving signal-to-noise ratio [26]. In SS-OCT a broadband swept source whose wavelength varies with time is used. Although SS-OCT acquisition speed is greater than that of SD-OCT (100 000 versus 85,000 A-scans/second) and its use of longer wavelengths (1060 nm versus 840–850 nm in SD-OCT) achieves greater tissue penetration, SD-OCT equipment is widely available and found in a large number of hospitals [27].

In conclusion, the study has potential, but it should be completed and revised for publication.

Author´s response:

Once again, thank you very much for your review, which will improve the quality of our manuscript.

Comments on the Quality of English Language

The English and style are in general fine throughout the manuscript. However, some issues exist, for example please avoid one-phrase paragraphs. The manuscript should be revised from the point of view of style.

Reviewer 3 Report

Comments and Suggestions for Authors

This study compared the structural differences produced by Alzheimer's disease and multiple sclerosis in the retina using optical coherence tomography. It is known that both diseases cause retinal thinning, while this study directly compares the two diseases in terms of the structural differences in the retina. This comparison might help to improve the understanding of the underlying mechanisms of these diseases and might lead to the development of new diagnostic and therapeutic approaches. There are some questions need to be addressed.

One of the main challenges of using optical coherence tomography (OCT) is obtaining high-quality images. OCT requires the patient to be able to fixate on a target for several seconds while the images are being taken, which can be difficult for some patients, particularly those with cognitive or visual impairments. In addition, OCT images can be affected by factors such as cataracts, vitreous opacities, and media opacities, which can make it difficult to obtain accurate measurements of retinal thickness. Another challenge is the interpretation of OCT images, which requires specialised training and expertise. Could the author answer how to overcome these limitations by using OCT to access these data?

Are there other imaging techniques that allow the measurements of retinal thickness, such as fundus photography, scanning laser ophthalmoscopy, and adaptive optics imaging? What is the advantage of OCT over these techniques?

Line 100 from which equation or modal to determine 12 eyes in each group?

There is a typo of table 1 first row, please check the age for the MS patients.

In figure 4 and table 2, there are two types of patients, MS and AD. Why is only one result of AUROC value? 

In figure 8, what is the unit of y axis? Some subfigure missing y labels.

Author Response

Comments and Suggestions for Authors

This study compared the structural differences produced by Alzheimer's disease and multiple sclerosis in the retina using optical coherence tomography. It is known that both diseases cause retinal thinning, while this study directly compares the two diseases in terms of the structural differences in the retina. This comparison might help to improve the understanding of the underlying mechanisms of these diseases and might lead to the development of new diagnostic and therapeutic approaches. There are some questions need to be addressed.

One of the main challenges of using optical coherence tomography (OCT) is obtaining high-quality images. OCT requires the patient to be able to fixate on a target for several seconds while the images are being taken, which can be difficult for some patients, particularly those with cognitive or visual impairments. In addition, OCT images can be affected by factors such as cataracts, vitreous opacities, and media opacities, which can make it difficult to obtain accurate measurements of retinal thickness. Another challenge is the interpretation of OCT images, which requires specialised training and expertise. Could the author answer how to overcome these limitations by using OCT to access these data?

Are there other imaging techniques that allow the measurements of retinal thickness, such as fundus photography, scanning laser ophthalmoscopy, and adaptive optics imaging? What is the advantage of OCT over these techniques?

To guarantee the correct performance of the exploratory protocol, we selected patients with a visual acuity greater than 0.5 on the Snellen scale in each eye. This ensured that they were able to fix their gaze and that they did not have large cataracts or significant media opacities. In any case, the exclusion criteria included significant opacities and media and ophthalmological or systemic pathologies that could influence the results of the study.

The OCT images were interpreted by the neuro-ophthalmologists on the research team, who have extensive experience in interpreting OCT images.

We have improved the limitations paragraph of the Discussion section to explain how we have controlled for these potential biases:

The main weakness of this study is the small database due to the difficulty of obtaining MS and AD recordings for age-matched populations. This is significant because it is important that both groups are similar in age so that there is no additional retinal layer loss due to ageing. Another potential limitation, controlled in this study through the inclusion and exclusion criteria, is the presence of ocular disturbances such as cataracts, which could prevent gaze fixation during the exploratory protocol. In this study, the OCT images were interpreted by two neuro-ophthalmologists with extensive experience in both this field and in assessing OCT tests.

Although there are other digital image analysis techniques that can analyse the retina, in particular scanning confocal laser measurement, the resolution of OCT is so high and it is so widely used that these other techniques are employed increasingly infrequently and are relegated solely to the study of glaucoma. The advantages of OCT over these other techniques are its higher resolution and sensitivity, its ease of use, its greater repeatability, and the facts that it is completely harmless and takes very little time to perform.

Line 100 from which equation or modal to determine 12 eyes in each group?

We have improved the explanation in paragraph 2 of Section 2.1 Study cohort:

Based on a preliminary study of MS patients conducted by our group [19], we computed the sample size needed to detect differences of at least 6 μm in OCT-measured thicknesses. We used a bilateral test with an α 5% risk and a β 10% risk, i.e. with a power of 90%. To obtain enough patients for an in-depth study of the natural history of MS, it was decided to have equal numbers of non-exposed and exposed patients (ratio of 0.5). Based on these calculations, it was concluded that at least 12 eyes were needed in each group. Standard clinical and neuroimaging criteria were used as a basis for definitive diagnosis of MS [4]. To ensure a homogeneous population, only patients with the RRMS phenotype and without a history of optic neuritis in either eye were included”.

There is a type of table 1 first row, please check the age for the MS patients.

Author´s response:

Thank you very much, the error has been corrected.

In figure 4 and table 2, there are two types of patients, MS and AD. Why is only one result of AUROC value? 

Author´s response:

Subsection 3.2 makes the comparison between control subjects and patients (diseased subjects: MS patients and AD patients). This situation is clarified by the following sentences:

3.2. Differences between control subjects and patients (MS and AD).

  • Figure 5 and Table 2 show the AUROC values for the 4 layers studied in the control subjects and patients (MS and AD).
  • Figure 5. AUROC values: controls vs patients (MS and AD).
  • Table 2. Controls vs patients (MS and AD).

In figure 8, what is the unit of y axis? Some subfigure missing y labels.

Author´s response:

Figure 9 (we have added a new figure in the revised manuscript) has been modified and the units are included on the y-axis.

Round 2

Reviewer 1 Report

Comments and Suggestions for Authors

Thank you for your detailed response.

We hope that this research will lead to future clinics and the next study.

Author Response

Thank you for your positive review of the manuscript

Reviewer 2 Report

Comments and Suggestions for Authors

The manuscript was well corrected according to all the comments and suggestions. However, as the authors stated, ”we can include examples of images that we discarded for these reasons as supplementary material”. Although there is a literature on this topic, it would be useful for the paper to have such a supplementary material added, with good OCT images, as well as with others that were discarded - for different reasons (i.e., with different scores). 

Also, it would be prudent (for authors and journal) to highlight in the last paragraph of the last section that this is an exploratory study regarding such an assessment, using thickness evaluation. Please remove the term ”ideal” (nothing is ideal...). Usually, in order to make a good diagnosis, there is a synergy of methods, therefore this idea must be highlighted at the end of Conclusions.

Apart from these aspects, in the opinion of this reviewer the paper could be accepted for publication.

Author Response

Thank you for the positive assessment of our manuscript.

We have added the following sentence in the last paragraph (conclusions) as you suggested and we have deleted the term "ideal":

"This is an exploratory study in relation to the diagnosis of two common diseases, using neuro-retinal thickness assessment by the innocuous OCT test. This test can be considered a useful technique that, in synergy with other diagnostic methods, helps to reach a definitive diagnosis earlier and more efficiently".

We have added the suplementary material with figures to explain examples of good OCT images, as well as with others that were discarded for different reasons (poor image quality, off-centre OCT image, artefacts). 
